# Elevated Plasma Apurinic/Apyrimidinic Endonuclease 1/Redox Effector Factor-1 Levels in Refractory Kawasaki Disease

**DOI:** 10.3390/biomedicines10010190

**Published:** 2022-01-17

**Authors:** Yu-Ran Lee, Eun Young Bae, Hong Ryang Kil, Byeong-Hwa Jeon, Geena Kim

**Affiliations:** 1Department of Physiology, College of Medicine, Chungnam National University, Daejeon 35015, Korea; lyr0913@gmail.com; 2Department of Pediatrics, College of Medicine, Chungnam National University Hospital, Chungnam National University, Daejeon 35015, Korea; pebble1217@hanmail.net (E.Y.B.); gilhongr@gmail.com (H.R.K.); 3Department of Pediatrics, College of Medicine, Chungnam National University Sejong Hospital, Chungnam National University, Sejong 30099, Korea

**Keywords:** apurinic/apyrimidinic endonuclease-1/redox factor-1, mucocutaneous lymph node syndrome, vasculitis

## Abstract

Kawasaki disease (KD) refers to systemic vasculitis of medium-sized vessels accompanied by fever. The multifunctional protein apurinic/apyrimidinic endonuclease-1/redox factor-1 (APE1/Ref-1) is a new biomarker for vascular inflammation. Here, we investigated the association between APE1/Ref-1 and KD. Three groups, including 32 patients with KD (KD group), 33 patients with fever (Fever group), and 19 healthy individuals (Healthy group), were prospectively analyzed. APE1/Ref-1 levels were measured, and the clinical characteristics of KD were evaluated. The mean age of all patients was 2.7 ± 1.8 years, but the Healthy group participants were older than the other participants. Fever duration was longer in the KD group than in the fever group. APE1/Ref-1 levels were significantly higher in the KD group (*p* = 0.004) than in the other two groups, but there was no difference between the healthy and fever groups. APE1/Ref-1 levels did not differ according to fever duration or coronary arterial lesion but were higher in refractory KD cases than in non-refractory cases. APE1/Ref-1 levels were significantly higher during the acute phase of KD. We propose that APE1/Ref-1 could be a beneficial biological marker for the diagnosis and prognosis of KD, especially in refractory KD.

## 1. Introduction

Kawasaki disease (KD) is an acute febrile disease diagnosed in young children or infants aged less than 6 years [1]. Kawasaki and Naoe reported KD in 1967 [2]. It is characterized by a persistent fever above 38 ℃ over 5 days, and its clinical symptoms include nonsuppurative bilateral conjunctival injection, red lips, strawberry tongue, atypical exanthema, cervical lymph node enlargement, swelling and erythema of hands and feet, and membranous desquamation [3]. Patients who meet the criteria according to principal clinical findings are considered as complete KD cases, and those who do not meet the criteria are diagnosed with incomplete KD [3]. Most KD cases are manageable by treatment, without any complications. Intravenous immune globulin is the standard treatment for the acute stage of KD [3]. However, this disease may be accompanied by diverse cardiovascular complications such as coronary aneurysms, heart failure in the acute complication stage, and myocardial infarction in patients with large coronary aneurysms, and these complications are more likely associated with refractory KD. Refractory KD means that the patients are resistant to the standard immunoglobulin treatment for KD. [4]. KD is a type of systemic vasculitis primarily involving the arterial wall [5]. Various KD biomarkers have been suggested for definitive KD diagnosis [6,7]. Apurinic/apyrimidinic endonuclease-1/redox factor-1 (APE1/Ref-1) is a multifunctional protein that plays roles in transcriptional regulation through redox modification and base excision repair [8]. It has been suggested that alterations in APE1/Ref-1 are associated with various diseases, including cancer, neurodegenerative disease, coronary arterial disease, murine myocarditis, and hypertension [9,10,11,12]. APE1/Ref-1 acts as a reductive activator of various transcription factors for controlling cell apoptosis, inflammation, and proliferation [12]. We hypothesized that APE1/Ref-1 could be a diagnostic biomarker for KD patients and may distinguish other patients with fever. The main objective of this study was to investigate the association between APE1/Ref-1 and KD and further analyze APE1/Ref-1 according to KD characteristics.

## 2. Materials and Methods

### 2.1. Patients and Data Collection

Patients with KD and fever and the healthy controls were prospectively enrolled from January 2020 to February 2021 at the pediatric department of Chungnam National University Hospital. The study was approved by the Chungnam National University Hospital Institutional Review Board, and informed consent forms were signed by the parents/legal guardians of the patients. The inclusion criteria for the KD group were admission and treatment of children with KD, for the fever group were admission as patients with fever, and for the healthy group were healthy patients from a pediatric outpatient clinic who visited for simple cardiac murmur or short stature. All the enrolled participants were >1 month and <8 years of age. Exclusion criteria included neonates, a severe infection that required intensive care, or a malignant condition accompanied by congenital heart disease. KD patients met the diagnostic criteria for KD established in 2017 by the American Heart Association [3]. Age at diagnosis, sex, weight, fever duration, presence of complete KD, findings from blood examination, responsiveness to immunoglobulin, the Kobayashi score, Egami score, Sano score, and echocardiography results were examined for children in the KD group [13]. All patients with KD underwent echocardiography at diagnosis, at 2 weeks, and at 2 months. The acute-phase coronary arterial lesion was determined based on echocardiography findings conducted 2 weeks after admission [3]. Patients with KD were treated with 2 g/kg intravenous immunoglobulin (IVIG) administered as a single infusion and medium-dose aspirin (50 mg/kg/day). Immunoglobulin was re-administered if persistent fever occurred at least 36 h after the IVIG infusion. A methylprednisolone pulse of 30 mg/kg and infliximab were sequentially administered when there was no response to immunoglobulin re-administration. Three days after the initial aspirin administration, the dose was reduced to 3–5 mg/kg/day. The fever group participants were evaluated for age, sex, weight, fever duration, and blood examination. Some patients were diagnosed with viral infections such as rhinovirus, adenovirus, acute pharyngitis, acute otitis media, acute gastroenteritis, acute bronchitis, or pneumonia in the fever group; they were given supportive care and antibiotics treatment according to diagnosis.

### 2.2. Measurement of APE1/Ref-1 Levels

All blood samples were collected in vacuum tubes during KD and fever diagnoses. Blood samples in the fever group were obtained at admission due to fever. The plasma was centrifuged at 3000× *g* rpm for 10 min to obtain cell-free samples. APE1/Ref-1 levels were determined using an APE1/Ref-1 sandwich enzyme-linked immunosorbent assay (ELISA) kit (MediRedox, Daejeon, Korea) according to the manufacturer’s instructions. Secreted APE1/Ref-1 levels in plasma (ng/mL) were calculated against a standard curve generated using recombinant human APE1/Ref-1 protein (MediRedox).

### 2.3. Statistical Analysis

SPSS Statistical Package version 21.0 (IBM Corp., Armonk, NY, USA) was used for all statistical analyses. Data are presented as mean ± standard deviation (SD), number (%), or mean value ± standard deviation for normally distributed data. Independent *t*-tests were used to compare continuous variables, while chi-square tests were used for comparing categorical variables. Variables between two groups were compared using an unpaired Student’s *t*-test, and the three groups were compared using a one-way ANOVA. The correlation between APE1/Ref-1 levels and other laboratory data was analyzed using Pearson’s correlation analysis. Statistical significance was set at *p* < 0.05.

## 3. Results

### 3.1. Patient Characteristics

Patient baseline characteristics are presented in Table 1. The mean age was 2.6 ± 1.6 years for the KD group, 2.3 ± 1.9 years for the fever group, and 3.6 ± 1.7 years for the healthy group participants. The age of the healthy group was significantly higher than that of the other two groups (*p* = 0.029). No difference was found in sex distribution or bodyweight among the three groups. Fever duration in the KD group was longer than that in the fever group, at 5.0 ± 1.6 d and 3.0 ± 2.8 d, respectively.

### 3.2. Laboratory Results

Laboratory results for the KD, fever, and healthy groups are shown in Table 2. White blood cell (WBC) counts, hemoglobin levels, platelet counts, number of segment neutrophils, and levels of aspartate aminotransferase (AST), alanine aminotransferase (ALT), protein, albumin, bilirubin, blood urea nitrogen (BUN), creatinine, sodium, and C-reactive protein (CRP) were analyzed. Only the KD group was evaluated for N-terminal pro-b-type natriuretic peptide (NT-proBNP) levels. As shown in Table 2, platelet count and total protein, bilirubin, BUN, and creatinine levels showed no significant difference among the three groups. WBC counts and the percentage of segment neutrophils were significantly higher in the KD and fever groups than in the healthy group (*p* < 0.001, *p* < 0.001, respectively). Hemoglobin levels were significantly different among the three groups (*p* = 0.003). The mean ± 2 SD of hemoglobin levels in the KD group was 11.0 ± 1.2 g/dL, which was lower than that in the fever group at 11.5 ± 1.0 g/dL (*p* = 0.041). The mean ± 2 SD of CRP level was higher in the KD group than in the fever group (8.5 ± 6.0 mg/dL vs. 3.4 ± 5.0 mg/dL, *p* < 0.001). Albumin showed differences among the three groups (*p* < 0.001). The mean ± 2 SD of albumin levels in the KD group was lower than that in the fever group (3.4 ± 0.4 g/dL vs. 4.0 ± 0.4 g/dL; *p* < 0.001). AST and ALT levels showed differences among the three groups (*p* = 0.039 and *p* < 0.001, respectively). AST levels were higher in the KD group than in the healthy group. ALT levels were higher in the KD group than in the fever group (139 ± 177 U/L vs. 27 ± 46 U/L; *p* = 0.001). Sodium levels differed among the three groups (*p* < 0.001), and the mean ± 2 SD of sodium levels in the KD group was higher than that in the fever group (135 ± 1 mEq/L, 137 ± 2 mEq/L, *p* = 0.007). The mean ± 2 SD of NT-proBNP in the KD group was 1315 ± 3345 pg/mL. We could not obtain the related data in the other two groups. However, a study that investigated the normal reference value of NT-proBNP in normal children reported that the median of NT-proBNP in 2–6-year-old children were 70 pg/mL (range 5–391 pg/mL) [14]. The mean ± 2 SD of APE1/Ref-1 levels was significantly higher in the KD group (0.654 ± 0.265 ng/mL than in the fever group (0.459 ± 0.290 ng/mL) and healthy group (0.442 ± 0.199 ng/mL), and the differences were significant (*p* = 0.004). Figure 1 shows the difference in APE1/Ref-1 levels among the three groups; the levels were higher in the KD group than in the fever group (*p* = 0.019) and the healthy group (*p* = 0.007). No difference was found in APE1/Ref-1 levels between groups 2 and 3. Additionally, the ROC curve of APE1/Ref-1 comparing KD and fever showed a cutoff level of 0.542 ng/mL for predicting KD, with an area under the curve of 0.682, a sensitivity of 60.6%, and specificity of 62.5%.

Next, we analyzed the correlation between APE1/Ref-1 levels and fever duration. No correlation was found. Additionally, there was no correlation between APE1/Ref-1 levels and other laboratory results, including hemoglobin, CRP, albumin, alanine aminotransferase, bilirubin, sodium, and NT-proBNP (Appendix A).

### 3.3. APE1/Ref-1 According to Characteristics of KD

Complete KD was observed in 13 of the 32 (40.6%) patients with KD, but there was no difference in APE1/Ref-1 levels relative to complete KD or incomplete KD. Four of the KD patients (12.5%) had coronary arterial lesions. Three patients with coronary arterial lesions were diagnosed at the time of diagnosis. One patient with coronary arterial lesion was diagnosed at 2 weeks of echocardiography. Yet, there was no difference in APE/Ref-1 levels based on the presence or absence of coronary arterial lesions. Predictive scores, i.e., the Kobayashi, Egami, and Sano scores, for responsiveness to immunoglobulins in KD were 2.4 ± 2.0, 1.8 ± 1.3, and 0.8 ± 0.8 (mean ± standard deviation), respectively. We further analyzed APE1/Ref-1 by dividing the patients into high-score and low-score groups and found no difference in the subgroups. Refractory KD that was nonresponsive to immunoglobulin was observed in 9 of the 32 cases of KD (28.1%). All the patients with refractory KD received methylprednisolone treatment, and none received infliximab. As shown in Figure 2, the mean level of APE1/Ref-1 in the refractory KD group was 0.803 ± 0.111 ng/mL compared to that of 0.459 ± 0.290 ng/mL in the fever group (*p* = 0.011) and 0.596 ± 0.044 ng/mL in non-refractory KD (*p* = 0.046). The ROC curve of APE1/Ref-1 comparing refractory vs. non-refractory and fever vs. refractory is presented in Figure 2.

## 4. Discussion

This study revealed the elevation of APE1/Ref-1 levels in patients with KD compared to that in patients with fever and healthy individuals. Patients with fever showed no difference in APE1/Ref-1 levels compared to healthy children. No correlation was found between APE1/Ref-1 levels and CRP or NT-proBNP, which are the established serum biomarkers of KD. Moreover, APE1/Ref-1 levels were higher in the refractory KD group than in the non-refractory KD and the fever group.

Many conditions, including cancer, hypertension, atherosclerosis, and particularly cardiovascular diseases, such as myocarditis and coronary artery disease, can lead to an elevation in APE1/Ref-1 levels [9,10,11,12,15,16]. APE1/Ref-1 levels are also elevated in response to radiation, reactive oxygen species (ROS), ischemia/reperfusion, and hypoxia. We found an association between the acute stage of KD and the point at which APE1/Ref-1 levels were elevated in conditions of vascular inflammation.

According to a study by Jin et al., the elevation in APE1/Ref-1 levels is associated with chronic inflammation, ROS, and ischemia/reperfusion injury in coronary artery disease and myocarditis [10]. These investigators also suggested that APE1/Ref-1 elevation is not a non-specific result due solely to inflammation, as they found no correlation with hsCRP. This is consistent with the findings of our current study in that our data also showed that APE1/Ref-1 was not correlated with CRP or NT-proBNP levels (Appendix A). The protective effect of APE1/Ref-1 is different from that of other inflammatory biomarkers or other cardiac biomarkers. The repair role of APE1/Ref-a is also suggested from findings of a murine myocarditis model in that APE1/Ref-1 is elevated at a later time point, in contrast to that of other markers, such as IL-1β, IFN-β, and IL-6 expression, which disappear with a decline in virus titer [11]. We believe these protective and repairable effects of APE1/Ref-1 were influenced by the prolonged elevation of APE1/Ref-1 after the acute stage of KD.

We hypothesize several mechanisms underlying APE1/Ref-1 elevation in KD. During the acute stage of KD, monocyte/macrophage-dominant inflammatory cell infiltration is observed throughout the entire arterial wall [17]. Activated inflammatory cells are highly stimulatory and orchestrate various reactions, including ROS production. Increased inducible nitric oxide synthase (iNOS) levels are observed in the infiltrating and accumulating inflammatory cells and the vascular smooth muscle cells, producing a large amount of nitric oxide (NO). The NO of iNOS origin is an unstable radical and forms peroxynitrite under oxidative stress, which leads to vascular damage [18].

In animal studies, plasma APE1/Ref-1 secretion might be strongly correlated with plasma NO levels induced by TNF-α or lipopolysaccharides [19]. In addition, recombinant human APE1/Ref-1 treatment suppresses TNF-α-induced VCAM-1 expression in endothelial cells, suggesting a functional role for extracellular APE1/Ref-1 [19]. TNF-α is an inflammatory mediator that is a potent activator of endothelial cells and is a key mediator in KD. An early study showed that TNF-α induces endothelial cell apoptosis in the blood serum of children with KD [20]. Furthermore, serum TNF-α levels are significantly elevated in children with acute KD, and they correlate with the incidence of coronary artery aneurysms [21]. In an animal model of KD induced by *Lactobacillus casei* cell wall extract, the process of coronary arteritis and aneurysms can be ablated by blocking the TNF receptor, suggesting that TNF-α can directly induce coronary artery lesions [22]. Various studies have also reported the clinical efficacy of blocking TNF-α production in children with refractory KD resistant to standard immunoglobulin and aspirin treatments. The fact that TNF-α is a key mediator of KD may be associated with the observation in the current study of APE1/Ref-1 being higher in refractory KD compared to that of non-refractory KD. We could not determine if the action of TNF-α induced the secretion of APE1/Ref-1 in refractory KD.

A major concern regarding KD is the diagnosis and treatment of refractory KD. Refractory KD presents with prolonged fever, eventually leading to a coronary arterial aneurysm [23]. New sensitive and specific biomarkers are needed for the diagnosis and prognosis of refractory KD. High APE1/Ref-1 levels in refractory KD observed in this study represent meaningful data. Although this study revealed that APE1/Ref-1 levels in patients with KD were higher than those in patients with fever and the healthy individuals, the non-refractory KD group did not show a significant difference in APE1/Ref-1 levels when compared with the fever group. However, considering that it is difficult to predict during diagnosis whether KD is refractory or non-refractory, and refractory KD cannot be determined in an earlier phase, the elevation of APE1/Ref-1 levels in KD compared to that in the fever group could be a helpful and important finding, which can be applied in the clinical setting. Elevated APE1/Ref-1 levels could be used to predict refractory KD and to distinguish between non-refractory and refractory KD.

The levels of APE1/Ref-1 in our study did not correlate with the presence of coronary arterial lesions, a KD complication. We obtained a sample during diagnosis. Although coronary arterial complications often begin during the first two weeks after fever onset, they can continue for months to years in a small subset of patients [3]. The small number of study participants with coronary arterial lesions among all patients is notable, and further study is needed in the future.

Our study did have several limitations. For instance, the study included a small number of patients, and there was a lack of data regarding serial time points of follow-up in KD patients. Thus, this was a cross-sectional measurement of KD at the point of diagnosis. It will be necessary to analyze serial data of APE1/Ref-1 in the long-term aspects of KD. However, follow-up research on KD may be complicated by aspirin being one of the treatments for KD as it could influence the APE1/Ref-1 levels. It is necessary to conduct a follow-up study. Using an animal model of KD would allow the exception of aspirin, providing insight into its effect. Another limitation is that there is no standard value for APE1/Ref-1 in children based on sex or other parameters.

In conclusion, APE1/Ref-1 levels were significantly higher in patients with KD during the acute phase of disease than in patients with fever and healthy children. Moreover, APE1/Ref-1 levels were significantly higher in patients with refractory KD. We suggest that APE1/Ref-1 could help diagnose KD, especially patients with refractory KD, in addition to improving its prognosis.

## Figures and Tables

**Figure 1 biomedicines-10-00190-f001:**
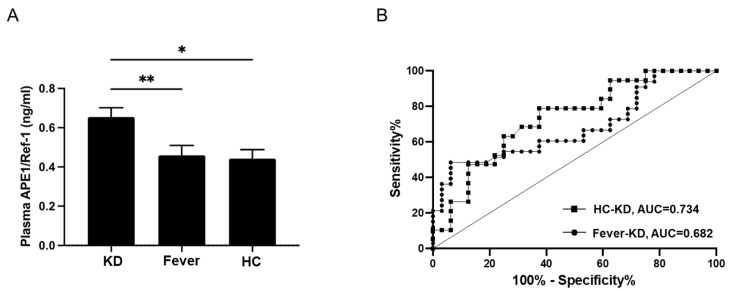
APE1/Ref-1 levels in the three groups, i.e., KD, fever, and healthy control groups. (**A**) APE1/Ref-1 levels were higher in the KD group than in the fever group (** *p* = 0.019) and the healthy control group (* *p* = 0.007). (**B**) The ROC curve of APE1/Ref-1 predicting KD, the cutoff value of APE1/Ref-1 predicting KD compared with that in the fever group was 0.542 ng/mL (AUC = 0.682; sensitivity of 60.6%; specificity of 62.5%), and the cutoff value of APE1/Ref-1 predicting KD compared with that in the healthy group was 0.482 ng/mL (AUC = 0.734; sensitivity of 68.4%; specificity of 68.7%), KD, Kawasaki disease; HC, healthy control.

**Figure 2 biomedicines-10-00190-f002:**
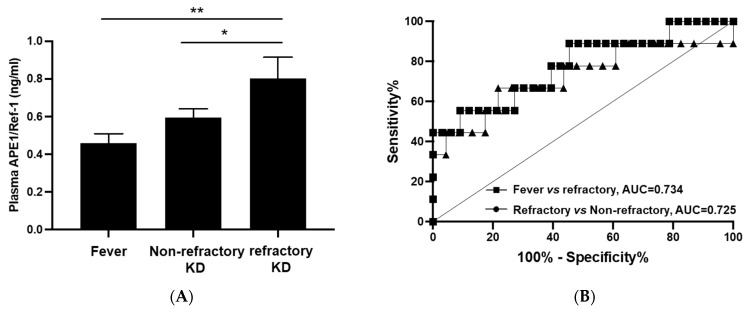
APE1/Ref-1 levels between the fever, non-refractory KD, and refractory KD groups. (**A**) APE1/Ref-1 levels between the fever and refractory KD groups showed a significant difference (** *p* = 0.011) and those between the non-refractory and refractory KD also showed a significant difference (* *p* = 0.046) (**B**) The ROC curve of APE1/Ref-1 predicting refractory KD, (Fever vs. refractory KD, AUC = 0.734, Refractory vs. non-refractory, AUC = 0.725).

**Table 1 biomedicines-10-00190-t001:** Patient baseline characteristics.

Variables	KD (n = 32)	Fever (n = 33)	Healthy Control (n = 19)	*p* Value
	n (%) or Mean ± SD	n (%) or Mean ± SD	n (%) or Mean ± SD	
Sex (male/female)	20 (62)	17 (51)	13 (68)	0.445
Age (years)	2.6 ± 1.6	2.3 ± 1.9	3.6 ± 1.7	0.029
Bodyweight (kg)	14.6 ± 5.4	12.7 ± 5.2	14.9 ± 4.5	0.235
Fever duration (days)	5.0 ± 1.6	3.0 ± 2.8		0.001

**Table 2 biomedicines-10-00190-t002:** Laboratory data.

Variables	KD Group (n = 32)	Fever Group (n = 33)	Healthy Group (n = 19)	*p* Value
	Mean ± SD	Mean ± SD	Mean ± SD	
WBC (/µL)	13,600 ± 3690	12,644 ±7201	7260 ± 1319	<0.001
Seg (%)	63 ±12	59 ± 17	40 ± 15	<0.001
Hb (g/dL)	11.0 ± 1.2	11.5 ± 1.0	12.1 ± 1.1	0.003
Platelet (×10^3^/µL)	336 ± 83	302 ± 113	331 ± 61	0.312
CRP (mg/dL)	8.5 ± 6.0	3.4 ± 5.0		<0.001
Total protein	6.4 ± 0.6	6.7 ± 0.6	6.5 ± 0.3	0.110
Albumin (g/dL)	3.4 ± 0.4	4.0 ± 0.4	4.2 ± 0.1	<0.001
AST (U/L)	98 ± 166	41 ± 23	30 ± 5	0.039
ALT (U/L)	139 ± 177	27 ± 46	15 ± 5	<0.001
Bilirubin (mg/dL)	0.7 ± 0.8	0.4 ± 0.2	0.5 ± 0.5	0.091
BUN (mg/dL)	10.0 ± 3.0	9.7 ±3.8	11.5 ±2.7	0.178
Creatinine (mg/dL)	0.25 ± 0.05	0.25 ± 0.09	0.30 ± 0.06	0.103
Sodium (mEq/L)	135 ± 1	137 ± 2	138 ± 1	<0.001
NT-proBNP (pg/mL)	1315 ± 3345			
APE1/Ref-1 (ng/mL)	0.654 ± 0.265	0.459 ± 0.290	0.442 ± 0.199	0.004

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
