# Peer review of "Elevated Plasma Apurinic/Apyrimidinic Endonuclease 1/Redox Effector Factor-1 Levels in Refractory Kawasaki Disease"

_biomedicines, 2022, doi:10.3390/biomedicines10010190_

Round 1

Reviewer 1 Report

This is an interesting study to reveal if APE1/Ref-1 is elevated in patients with Kawasaki disease compared to healthy controls or fever controls. In particular, it is interesting to note that APE1/Ref-1 may be elevated in refractory Kawasaki disease.

This is a well-written paper with good discussion, but I would like to see improvement in the following points.

At the end of the Introduction, the hypothesis is stated, but there is no statement of purpose. It is better to state the purpose clearly.

It seems that there is no need to redo the study, but it is questionable whether the choice of controls was appropriate. I believe that you should have selected febrile patients and healthy control children who were matched for age and sex with Kawasaki disease patients. In fact, the healthy control group was older but had the same weight as the Kawasaki disease and fever patients. (The healthy control group may have had growth retardation.)

In addition, the timing of blood collection for APE1/Ref-1 may have been different between patients with Kawasaki disease and those with fever control. I believe that the blood samples were taken 4-5 days after fever in the Kawasaki disease patients and much earlier than that in the fever control group. It would be better to specify the time of blood collection for the fever control group in the "Methods" section. If necessary, it should also be stated in Limitation.

Author Response

This is an interesting study to reveal if APE1/Ref-1 is elevated in patients with Kawasaki disease compared to healthy controls or fever controls. In particular, it is interesting to note that APE1/Ref-1 may be elevated in refractory Kawasaki disease.

This is a well-written paper with good discussion, but I would like to see improvement in the following points.

At the end of the Introduction, the hypothesis is stated, but there is no statement of purpose. It is better to state the purpose clearly.

         We have added a clear purpose of statement in the introduction section (lines 60-61).

It seems that there is no need to redo the study, but it is questionable whether the choice of controls was appropriate. I believe that you should have selected febrile patients and healthy control children who were matched for age and sex with Kawasaki disease patients. In fact, the healthy control group was older but had the same weight as the Kawasaki disease and fever patients. (The healthy control group may have had growth retardation.)

         I agree with you; All the enrolled participants were >1 month and <8 years of age. Short stature and growth retardation could be included in the healthy group.

In addition, the timing of blood collection for APE1/Ref-1 may have been different between patients with Kawasaki disease and those with fever control. I believe that the blood samples were taken 4-5 days after fever in the Kawasaki disease patients and much earlier than that in the fever control group. It would be better to specify the time of blood collection for the fever control group in the "Methods" section. If necessary, it should also be stated in Limitation.

         Per your suggestion, we have described the time of blood collection for the fever control group in the Methods section (lines 97-98).

Reviewer 2 Report

In this study, Dr. Lee and colleagues investigated the potential application of APE1/REF-1 as a specific biomarker for Kawasaki disease (KD). They performed a prospective study of 84 children divided into 3 groups: KD (32), Fever (33) and HC (19). They concluded that APE1/Ref-1 levels were significantly higher during the acute phase of KD and proposed that APE1/Ref-1 could be a beneficial biological marker for the diagnosis and prognosis of KD. Overall, this is a clinically relevant study. However, some major issues were found and need to be addressed by the authors:

  1. I would suggest a language evaluation by a native English writer. The manuscript contains too many typos and some words are not suitable. They unfortunately impair the readability of this manuscript. I will provide some examples below:
    • Line 32: what does it mean by "characteristically found from"? Please rephrase
    • "these complications are more likely present in refractory KD which was unresponsive intravenous immunoglobulin as usual management in KD" this is not grammatically correct. Please revise.
    • Line 50: instead of "exclude", maybe "distinguish" would be more appropriate
    • Lines 58, 59, 63, 67, 76, 97, 108 and so on, some words were typed twice. Please correct.
    • Line 64: if the "AHA" is not going to be used below this line, no need to abbreviate. Please remove
    • "Refractory KD that was nonresponsive to immunoglobulin was observed in 9 of the 32 cases of KD (28.1%) with a difference in APE1/Ref-1 levels occurring according to refractory KD or non-refractory KD." Please rephrase this sentence.
  2. Apart from the writing issue, there is a major problem requiring clarifications. The authors said "No correlation was found with APE1/Ref-1 and CRP or NT-proBNP," which part of the result backed this statement? It was clearly shown in Table 2 that CRP was significantly increased in KD than in fever group, similar with the APE1/REF-1. Moreover, there was no comparison of the NT-proBNP with anything else in that table. The authors need to clarify this discrepancy. 
  3. Also, healthy lab data need to be included in Table 2. Please provide.
  4. If the data is not available, at least the authors need to also compare (and analyze) the data of the KD and fever with normal cut-off values of healthy children at comparable age with the HC.
  5. The authors showed the APE1/REF-1 data for HC in Figure 1 (bar chart) but did not display the actual value in Table 2. Please add.
  6. Another striking finding is that the non-refractory KD has a non-significant increase of APE1/REF-1 as compared with fever group (Figure 2), which indicates that this inflammation marker is not sensitive to distinguish non-refractory KD from a regular fever. It could be that when the authors mixed the non-refractory and refractory KD into one group, the refractory APE1/REF-1 dominated the mean values. This is concerning because the non-refractory KD accounts for most of KD cases in general population and this suggests that the proposed biomarker is not good enough. The authors need to rephrase all the text including the title to indicate this unfortunate results, that APE1/REF-1 could only differentiate refractory KD. Otherwise, it will be a false claim.
  7. Line 37: Please add a sentence about the treatment of choice of KD. The authors has already mentioned it but it would be clearer to put it after this sentence.
  8. "However, this disease may accompany diverse cardiovascular complications such as coronary aneurysm, heart failure, and myocardial infarction" This needs a clarification whether those diseases are the complications of KD or they coexist or they are actually associated (influencing each other progression).
  9. Line 48: "Based on the fact that APE1/Ref-1 is altered in vascular inflammation..." Please elaborate this sentence. Which fact? Provide some citations as well.
  10. In the methods, the authors said "All enrolled participants were > 1 mo and < 8 yr in age" However, in table 1, none of the 3 groups reflect a child with age close to 8 years old. Is this correct? Please clarify.
  11. "Once values were determined, we then analyzed the correlation..." which values? Please specify.
  12. Line 142 onward, there is "complete" and "incomplete" KD, but there was no explanation about what they are. Please add their definition and criteria in the introduction.
  13. Line 144, "Four of the KD patients (12.5%) had coronary arterial lesions,..." please clarify whether this lesion was observed 2 weeks after diagnosis?
  14. I don't see the point of having Table 3 since it is not comparing anything. I think those values only need to be declared in the text.
  15. As I mentioned above, "An important finding of this study was the elevation of APE1/Ref-1 in patients with KD compared to that in patients with fever and healthy individuals" this sentence is not entirely true because only the refractory KD has a significant elevation of APE1/REF-1, but not the non-refractory. Please adapt the whole manuscript.
  16. "This is consistent with the findings of our current study in that our data also showed that APE1/Ref-1 was not correlated with CRP or NT-proBNP levels." Until the authors show the proof of no correlation, this is an inaccurate claim. 

Author Response

In this study, Dr. Lee and colleagues investigated the potential application of APE1/REF-1 as a specific biomarker for Kawasaki disease (KD). They performed a prospective study of 84 children divided into 3 groups: KD (32), Fever (33) and HC (19). They concluded that APE1/Ref-1 levels were significantly higher during the acute phase of KD and proposed that APE1/Ref-1 could be a beneficial biological marker for the diagnosis and prognosis of KD. Overall, this is a clinically relevant study. However, some major issues were found and need to be addressed by the authors:

  1. I would suggest a language evaluation by a native English writer. The manuscript contains too many typos and some words are not suitable. They unfortunately impair the readability of this manuscript. I will provide some examples below:
    • Line 32: what does it mean by "characteristically found from"? Please rephrase

à We have rephrased this part for better clarity.

    • "these complications are more likely present in refractory KD which was unresponsive intravenous immunoglobulin as usual management in KD" this is not grammatically correct. Please revise.

à Thank you for your comment; we have revised the sentence as follows:

“Refractory KD means that the patients are resistant to the standard immunoglobulin treatment for KD.”

    • Line 50: instead of "exclude", maybe "distinguish" would be more appropriate

à We have made the change as suggested.

    • Lines 58, 59, 63, 67, 76, 97, 108 and so on, some words were typed twice. Please correct.

à We have made the change as suggested.

Line 64: if the "AHA" is not going to be used below this line, no need to abbreviate. Please remove

à We have made the change as suggested.

    • "Refractory KD that was nonresponsive to immunoglobulin was observed in 9 of the 32 cases of KD (28.1%) with a difference in APE1/Ref-1 levels occurring according to refractory KD or non-refractory KD." Please rephrase this sentence.

à We have made the change as suggested.

  1. Apart from the writing issue, there is a major problem requiring clarifications. The authors said "No correlation was found with APE1/Ref-1 and CRP or NT-proBNP," which part of the result backed this statement? It was clearly shown in Table 2 that CRP was significantly increased in KD than in fever group, similar with the APE1/REF-1. Moreover, there was no comparison of the NT-proBNP with anything else in that table. The authors need to clarify this discrepancy. 

-à Thank you for this useful comment. Accordingly, we analyzed the correlation coefficient between APE1 and CRP or NT-pro-BNP and added this information as supplementary figure 1. Additionally, we have cited “supplementary figure 1” in lines 164.

Supplementary figure 1

Supplementary figure 1. Correlation between plasma APE1/Ref-1 levels and Kawasaki disease (coronary artery) marker. (A) Correlation between APE1/Ref-1 and CRP levels (B) correlation between APE1/Ref-1 and NT-proBNP levels. Statistical significance was determined via Pearson’s correlation analysis

  1. Also, healthy lab data need to be included in Table 2. Please provide.

à This has been corrected.

  1. If the data is not available, at least the authors need to also compare (and analyze) the data of the KD and fever with normal cut-off values of healthy children at comparable age with the HC.

à Laboratory data of the healthy group have been added in table 2.

  1. The authors showed the APE1/REF-1 data for HC in Figure 1 (bar chart) but did not display the actual value in Table 2. Please add.

à We have added this information.

  1. Another striking finding is that the non-refractory KD has a non-significant increase of APE1/REF-1 as compared with fever group (Figure 2), which indicates that this inflammation marker is not sensitive to distinguish non-refractory KD from a regular fever. It could be that when the authors mixed the non-refractory and refractory KD into one group, the refractory APE1/REF-1 dominated the mean values. This is concerning because the non-refractory KD accounts for most of KD cases in general population and this suggests that the proposed biomarker is not good enough. The authors need to rephrase all the text including the title to indicate this unfortunate results, that APE1/REF-1 could only differentiate refractory KD. Otherwise, it will be a false claim.

à Thank you for your sharp point. I agree with your opinion.

It is impossible to diagnose refractory or non-refractory KD in the acute stage of KD; it is later presented as a type of KD (non-refractory /refractory KD) after KD treatment. So, it could also be meaningful if the results show a significant difference in APE/Ref-1 between the KD and fever groups. Especially, it is important to diagnose KD early to reduce the number of fever days.

As evident from the ROC curve in Figure 2, it is possible to diagnose KD using the cut-off value of APE1/Ref-1 predicting KD. Per your comment, we corrected the text in the title, abstract, and discussion.

  1. Line 37: Please add a sentence about the treatment of choice of KD. The authors has already mentioned it but it would be clearer to put it after this sentence.

à We have revised the text accordingly.

  1. "However, this disease may accompany diverse cardiovascular complications such as coronary aneurysm, heart failure, and myocardial infarction" This needs a clarification whether those diseases are the complications of KD or they coexist or they are actually associated (influencing each other progression).

à For better clarity, we have revised this sentence as follows: However, this disease may accompany diverse cardiovascular complications such as coronary aneurysm, heart failure that could occur in acute stage complication, and myo-cardial infarction that could occur in patients with large coronary aneurysm

  1. Line 48: "Based on the fact that APE1/Ref-1 is altered in vascular inflammation..." Please elaborate this sentence. Which fact? Provide some citations as well.

à  We have revised this sentence for better clarity.

  1. APE1/Ref-1 is known to act as a reductive activator of many transcription factors in controlling apoptosis, inflammation, proliferation in cellular process (12). In the methods, the authors said "All enrolled participants were > 1 mo and < 8 yr in age" However, in table 1, none of the 3 groups reflect a child with age close to 8 years old. Is this correct? Please clarify.

à Thank you for your comment, We planned to enroll patients under the age of 8 years, but the enrolled patients were relatively younger. The data reflect the common age in KD (2-3 years), and there were also many younger patients in the fever group.

  1. "Once values were determined, we then analyzed the correlation..." which values? Please specify.

à We have corrected this sentence as follows: (In line 160).

“We analyzed the correlation between APE1/Ref-1 levels and fever duration.”

  1. Line 142 onward, there is "complete" and "incomplete" D, but there was no explanation about what they are. Please add their definition and criteria in the introduction.

à Thank you for your comment. We have revised this sentence (line 39-41) in the introduction.

  1. Line 144, "Four of the KD patients (12.5%) had coronary arterial lesions,..." please clarify whether this lesion was observed 2 weeks after diagnosis?

à Thank you for your comment, We have added sentence in line 180  in page 8,

All patients with coronary arterial lesion were diagnosed at 2 weeks of echocardiography.

  1. Four of the KD patients (12.5%) had coronary arterial lesions, and 3 patients with cor-onary arterial lesion were diagnosed at diagnosis and 1 patients with coronary arterial lesion were diagnosed at 2 weeks point of echocardiography.I don't see the point of having Table 3 since it is not comparing anything. I think those values only need to be declared in the text.

à Thank you for the comment. We agree with your opinion.

Accordingly, we removed table 3 and corrected the sentence in lines 183-185, as follows:

Predictive scores, i.e., the Kobayashi, Egami, and Sano scores, for responsiveness to immunoglobulins in KD were 2.4 ± 2.0, 1.8 ± 1.3, and 0.8 ± 0.8 (mean ± standard deviation), respectively.

  1. As I mentioned above, "An important finding of this study was the elevation of APE1/Ref-1 in patients with KD compared to that in patients with fever and healthy individuals" this sentence is not entirely true because only the refractory KD has a significant elevation of APE1/REF-1, but not the non-refractory. Please adapt the whole manuscript.

à We agree with your point. We revised the discussion to emphasize a significant elevation in “APE1/Ref-1 in refractory KD. In the discussion, the following text was added (lines 265-274):

: Although this study revealed that the elevation of APE1/Ref-1 levels in patients with KD was comparable to that in patients with fever and the healthy individuals, the non-refractory KD group did not show a significant difference in APE1/Ref-1 levels when compared with the fever group. Considering that many KD patients were non-refractory KD cases, it could be a limitation that APE1/Ref-1 is a specific biomarker for all types of KD. However, considering that it is difficult to predict during diagnosis whether KD is refractory or non-refractory, and refractory KD cannot not be determined in an earlier phase, the elevation of APE1/Ref-1 levels in KD compared to that in the fever group could be a helpful and important finding, which can applied in the clinical setting.

” Additionally, we have listed the study limitations according to your suggestion. (lines 289-290)

“Small portion of refractory KD in entire KD group could be a limiting factor for interpreting for difference of APE1/Ref-1 between KD group and fever group.”

  1. "This is consistent with the findings of our current study in that our data also showed that APE1/Ref-1 was not correlated with CRP or NT-proBNP levels." Until the authors show the proof of no correlation, this is an inaccurate claim. 

à Thank you for the detailed comment. As per your comment, we analyzed the correlation coefficient between APE1 and CRP or NT-pro-BNP and added the information as supplementary figure 1. We have cited “supplementary figure 1” in lines  223. 

Reviewer 3 Report

This is overall a nicely executed study. However, there are some shortcomings that I would wish to bring to the authors' attention, as follows:

  1. The goals of the study are poorly defined in the Introduction section. It should be highlighted why authors think APE1/Ref-1 would be a specific biomarker for Kawasaki's disease.
  2. What is the biological metabolism of APE1/Ref-1? How does it clear from the body? Is it dependent on renal function, hepatic function, body mass index? These important aspects should be considered in order to fully interpret obtained findings.
  3. I would wish to see logistic regression in the whole population (KD+fever+control group) in which the main dependent binary outcome would be Kawasaki disease (NO-0, YES-1) and then APE1/Ref-1 would be tested as an independent covariate against this variable adjusted for variables of age, sex, BMI, CRP, and NT-proBNP. Then authors should report multivariate-adjusted odds ratios (coefficient) for APE1/Ref-1 variable.
  4. Renal function parameters should be shown in the baseline characteristics/laboratory data table.
  5. Treatments should be presented in two compared columns, for example, KD group vs. fever group? How similar were these patients managed according to drug management of fever and underlying clinical presentation? This should be elaborated.

Author Response

This is overall a nicely executed study. However, there are some shortcomings that I would wish to bring to the authors' attention, as follows:

  1. The goals of the study are poorly defined in the Introduction section. It should be highlighted why authors think APE1/Ref-1 would be a specific biomarker for Kawasaki's disease.
  • Thank you for your comment, I agree with your opinion.

We have revised the introduction section to enhance clarity.

  1. What is the biological metabolism of APE1/Ref-1? How does it clear from the body? Is it dependent on renal function, hepatic function, body mass index? These important aspects should be considered in order to fully interpret obtained findings.

-> Thanks for your valuable comment.

Currently, research for specific biomarkers through the investigation of the correlation between APE1/Ref-1 in biological fluids  and certain diseases are being mainly conducted.

The biological metabolism of APE1/Ref-1 is not well known. APE1/Ref-1 is detected in urine as MW is 36.5 kDa. However, it is not clear whether APE1/Ref-1 in the blood is filtered and cleared. (Dis Markers, 2016;2016:7276502.)

Autophagy appears to play an important role in the degradation of APE1/Ref-1. Ectopically expressed APE1/Ref-1 was degraded by Parkin and PINK1 via polyubiquitination in mouse embryonic fibroblast cells. (Mol Carcinog, 2017, 56(2), 325-336). Recently, it has been reported that autophagy plays an important role in the degradation of the APE1/Ref-1. (Nature Commm, 2021, 12:16).

Unfortunately, there have been no studies on the clearance pathway for APE1/Ref-1 that can affect the plasma concentration of APE1/Ref-1.  It is undoubtedly an important part for further studies on the clearance of APE1/Ref-1.

  1. I would wish to see logistic regression in the whole population (KD+fever+control group) in which the main dependent binary outcome would be Kawasaki disease (NO-0, YES-1) and then APE1/Ref-1 would be tested as an independent covariate against this variable adjusted for variables of age, sex, BMI, CRP, and NT-proBNP. Then authors should report multivariate-adjusted odds ratios (coefficient) for APE1/Ref-1 variable.

à Thanks for this useful comment. We set KD disease to 1, and fever and healthy control to 0 using the APE1/Ref-1 value of the whole population (KD+Fever+Healthy control), analyzed via logistic regression, and generated an ROC curve. As shown in the analysis graph below, the AUC value of the ROC curve was 0.70, and the P-value was 0.002, which represented a significant result.

  1. Renal function parameters should be shown in the baseline characteristics/laboratory data table.

à Thank you for your advice. We have added the parameters of renal function in table 2.

  1. Treatments should be presented in two compared columns, for example, KD group vs. fever group? How similar were these patients managed according to drug management of fever and underlying clinical presentation? This should be elaborated.

à Thank you for this useful comment. We agree with your advice.

 We have described the treatment protocol in the methods section (lines 83-89) . In the KD groups, all nine refractory KD patients received the prednisolone pulse therapy after immunoglobulin therapy, and none received infliximab. Some patients were diagnosed with viral infections such as rhinovirus, adenovirus infection, acute pharyngitis, acute otitis media, acute gastroenteritis, acute bronchitis, or pneumonia in the fever group;  these patients were given supportive care and treated with antibiotics (acute otitis media).

As these data are too small to create a new table, we added the explanation about KD patients in lines 188-190 in 3.3. section. Additionally, we added information about the treatment in the Fever group, in lines 90-94 in section 2.1.

Round 2

Reviewer 2 Report

Thank you for the responsive answers to my previous comments. In the revised manuscript, I found some texts requiring clarifications:

  • "Although this study revealed that the elevation of APE1/Ref-1 levels in patients with KD was comparable to that in patients with fever and the healthy individuals, the non-refractory KD group did not show a significant difference in APE1/Ref-1 levels when compared with the fever group." This statement is incorrect according to Table 2 and Figure 1. Which data showed comparable APE1/Ref-1 in KD vs. fever and HC? Wasn't it higher (0.654 vs. 0.459; p 0.004)?
  • "Considering that many KD patients were non-refractory KD cases, it could be a limitation that APE1/Ref-1 is a specific biomarker for all types of KD.", which findings backed up this statement that APE1 is a biomarker for all KD types?
  • "A small portion of refractory KD in the entire KD group could be a limiting factor for interpreting the difference in APE1/Ref-1 levels between the KD group and the fever group." Is this correct? I think there has been a clear difference between KD and fever, but not non-refractory KD and fever. I am not sure how the portion of refractory KD influences such a non-significance?
  • Since the NT-proBNP was not measured in other groups, please compare it with the standard cut-off value in general population with the same age range as the KD group. It would be insightful to know if the value is higher than normal or not.

Author Response

  • "Although this study revealed that the elevation of APE1/Ref-1 levels in patients with KD was comparable to that in patients with fever and the healthy individuals, the non-refractory KD group did not show a significant difference in APE1/Ref-1 levels when compared with the fever group." This statement is incorrect according to Table 2 and Figure 1. Which data showed comparable APE1/Ref-1 in KD vs. fever and HC? Wasn't it higher (0.654 vs. 0.459; p 0.004)?

--> Thank you for your comment.

I have corrected the sentence in lines 237-239.

Although this study revealed that APE1/Ref-1 levels in patients with KD were higher than those in patients with fever and the healthy individuals,

  • "Considering that many KD patients were non-refractory KD cases, it could be a limitation that APE1/Ref-1 is a specific biomarker for all types of KD.", which findings backed up this statement that APE1 is a biomarker for all KD types?

--> Thank you for your comment.

I have corrected the sentence as follows:

The comparable difference between the refractory KD and fever groups could be due to the difference in APE1/Ref-1 levels between the two groups (lines 240-242).

  • "A small portion of refractory KD in the entire KD group could be a limiting factor for interpreting the difference in APE1/Ref-1 levels between the KD group and the fever group." Is this correct? I think there has been a clear difference between KD and fever, but not non-refractoryKD and fever. I am not sure how the portion of refractory KD influences such a non-significance?

--> Thank you for your comment.

I have removed the indicated sentence.

  • Since the NT-proBNP was not measured in other groups, please compare it with the standard cut-off value in general population with the same age range as the KD group. It would be insightful to know if the value is higher than normal or not.

-->  Thank you for your comment.

I have added the following sentences in lines 133-137.

The mean ± 2 SD of NT-proBNP in the KD group was 1315 ± 3345 pg/mL. We could not obtain the related data in the other two groups. However, a study that investigated the normal reference value of NT-proBNP in normal children reported that the median of NT-proBNP in 2‒6 year old children was 70 pg/mL(range 5‒391 pg/mL) (14).  

Reviewer 3 Report

The authors have answered all my queries satisfactorily. 

Author Response

Thank you for your comment.

Round 3

Reviewer 2 Report

Regarding this statement “The comparable difference between the refractory KD and fever groups could be due to the difference in APE1/Ref-1 levels between the two groups” again, this is not true. Refractory KD had a significantly higher APE1 than fever group, so not sure what was the basis of this sentence. Also, comparable difference of what? What did the authors want to say in this paragraph?

FYI, comparable means similar in size, amount, or quality to something else (Cambridge Dictionary). 

I think the authors definitely need a help from a native English scientist to really read and revise the text not only grammatically but also for clarity, coherence and readability. This person has to be a person who knows about this study, to avoid doing similar false claim again, and again.

Author Response

  • Thank you for the valuable comment. To avoid misunderstanding, some sentences that may cause confusion have been deleted. Notably, sentences beginning with "comparable difference" were deleted as they could confuse readers. For example, we modified the above-mentioned sentence for improved readability and clarity as follows in lines 245-247: "Elevated APE1/Ref-1 levels could be used to predict refractory KD and distinguish between non-refractory and refractory KD.” Once again, we apologize for any confusion caused by the difficult-to-understand expressions.

Round 4

Reviewer 2 Report

Thank you for the corrections. I have no remaining remarks.